# Urinary Tract Infections in Kidney Transplant Patients: An Open Challenge—Update on Epidemiology, Risk Factors and Management

**DOI:** 10.3390/microorganisms12112217

**Published:** 2024-10-31

**Authors:** Biagio Pinchera, Emilia Trucillo, Alessia D’Agostino, Ivan Gentile

**Affiliations:** Department of Clinical Medicine and Surgery, Section of Infectious Diseases, University of Naples “Federico II”, Via Sergio Pansini 5, 80131 Naples, Italy; emiliatrucillo@libero.it (E.T.); alessiadagostino53@gmail.com (A.D.); ivan.gentile@unina.it (I.G.)

**Keywords:** urinary tract infections, kidney transplant patients, SOT, MDR, epidemiology, risk factors, management

## Abstract

Urinary tract infections are one of the main complications in kidney transplant patients, with a significant impact on graft function and survival. In fact, it is estimated that up to 74% of kidney transplant patients experience at least one episode of UTIs in the first year after transplantation, with an increased risk of graft loss and an increased risk of mortality. Several risk factors have been identified, such as female gender, old age, diabetes mellitus, immunosuppression, pre-transplant UTIs, urinary tract abnormalities, and prolonged dialysis. The worsening burden of antimicrobial resistance is also in itself a risk factor and a major complication in evolution and management. The management of prophylaxis, asymptomatic bacteriuria, and UTIs is still an open challenge, with some points to be clarified. Faced with such scenarios, our review aimed to evaluate the current epidemiology, examine the risk factors, and consider all the possibilities and methods of management, giving a current view and evaluation of the topic.

## 1. Introduction

In kidney transplant recipients, infectious complications and, in particular, Urinary Tract Infections (UTIs) are very common and are associated with a high risk of graft loss and a high impact on survival.

Our review aimed to evaluate the current epidemiology, examine the risk factors, and consider all the possibilities and methods of management, giving a current view and evaluation of the topic.

## 2. Epidemiology

UTIs are the most common type of infection in kidney transplant recipients. The incidence rate ranges from 28% to 38% over the course of a lifetime. The frequency of recurrent UTIs varies from 14% to 44% [1,2].

UTIs can occur at any time after transplantation, but they are mainly observed in the first 3–6 months [3,4].

In addition, data in the literature show that the incidence of infections is higher in the early post-transplant period, especially in the first year (74%) [4], and it decreases to 35% during the second year and to 21% at four years after transplantation [5]. The high risk of UTI development in the early post-transplant period is the result of the interaction of several factors, such as surgical trauma, catheter and ureteral stent placement, and high levels of immunosuppression.

UTIs represent the main cause of sepsis during the early post-transplant period [6].

Moreover, the incidence of UTIs appears to be higher in kidney transplant patients than in other solid organ transplant recipients, as highlighted by the RESISTRA study, a multi-center prospective cohort study that enrolled 4388 solid organ transplant (SOT) recipients for follow-up for one year. The study highlighted 249 episodes of UTIs in 192 recipients and bacterial UTIs (82.5% episodes of cystitis and 17.4% of pyelonephritis), which occurred in 150 kidney transplants (7.3%) and in 6 kidney–pancreas transplant recipients (4.9%). All cases of pyelonephritis were observed in the kidney transplant population. The recurrence of the infection has been described in 14.7% (22/150) of kidney transplant patients and has not been detected in kidney–pancreas transplant recipients. The highest incidence rate of UTIs in this setting was observed in the first six months [7,8,9].

Similarly, in a retrospective study conducted at the emergency department, it was estimated that kidney transplant patients were hospitalized over three times more often (kidney 66%, kidney/pancreas 15%, liver 13%, heart 3%, lung 3%). UTIs were the most common cause of admission (43% of cases). Nine of 77 patients (11.7%) developed sepsis [6].

## 3. Microbiology

UTIs in kidney transplant recipients are usually caused by Gram-negative bacteria (>70%), and *Escherichia coli* is the most frequently detected pathogen (30–80%) [5,10].

Other common bugs are *Klebsiella*, *Pseudomonas aeruginosa* and *Proteus*. There is also a growing increase in antibiotic resistance due at least partially to the widespread prescription of antibiotic therapies for asymptomatic bacteriuria and for the prevention of infection, such as cotrimoxazole (TMP-SMX) and fluoroquinolones. The presence of multi-drug-resistant (MDR) and extensively-drug-resistant (XDR) pathogens is increasing.

Gram-positive pathogens (*Streptococcus* species, *Staphylococcus saprophyticus*) less frequently cause UTIs. Moreover, among fungal infections, Candida species are the most frequent etiological agents in UTIs in kidney transplant recipients, occurring in about 11% of cases [11]. Detection of *Staphylococcus aureus* on urine culture can be related to hematogenous spread (i.e., from bacteremia) rather than an ascending infection from the urinary tract. Not all microorganisms detected in urine samples represent pathogens, such as *Staphylococcus epidermidis* (except in the presence of ureteral stents), *Lactobacillus*, and *Gardnerella vaginalis*. *Corynebacterium urealyticum* may represent an important pathogen potentially associated with obstructive uropathy and/or cystitis [12].

With regard to antibiotic resistance, *E. coli* produced extended-spectrum b-lactamase (ESBL) in 26.3% of cases (31/118). Resistance to quinolones was detected in 38% of *E. coli*, 31% of *Klebsiella* species, and 21% of *P. aeruginosa*. Moreover, 77% of *E. coli*, 81% of *Klebsiella* species, and 80% of *P. aeruginosa* showed resistance to cotrimoxazole. No resistance to vancomycin was detected in *Enterococcus* species isolates [7].US National Health Safety Network surveillance data from 2014 reported resistance to carbapenems in 9.5% of *Klebsiella* spp., 1.1% of *E. coli*, and 23.9% of *Pseudomonas aeruginosa* isolates causing catheter-associated UTIs [13]. According to the current data in the literature, *K. pneumoniae* seems to be the second most common pathogen causing UTIs in kidney transplant patients, following *Escherichia coli* [14].

In a retrospective study with 335 kidney transplant recipients at Gdansk Transplantation Centre during a follow-up period of 12 months, 59 *Klebsiella* spp. UTIs were recorded in 24 patients. The 59 cases included 30 episodes of asymptomatic bacteriuria (AB), 11 cases of lower UTI, 10 episodes of pyelonephritis (AGPN), and 8 cases of urosepsis. More than 60% of these episodes were caused by ESBL+, whereas no carbapenemase-producing strains were detected. Almost 80% of UTIs were observed from the second month post-transplantation. ESBL-producing strains were most often detected in upper *Klebsiella* spp. UTIs were responsible for 62.5% of cases of urosepsis [15].

In a retrospective study by Nihan Tekkarışmaz et al., conducted in Turkey, 145 patients who received kidney transplants between 2010 and 2017 were enrolled and evaluated. All patients received prophylaxis for *Pneumocystis carinii* with cotrimoxazole (160/800 mg/day) for the first 6 months. The rate of UTIs was found to be 37.9%. Asymptomatic bacteriuria, uncomplicated UTIs, and complicated UTIs were observed in 5.4%, 58.2%, and 36.4%, respectively (urosepsis in 9% of episodes). Recurrent UTIs occurred in 32.7% of the patients. *Escherichia coli* (n = 43, 41%) was the most common microorganism detected. Other pathogens in order of frequency were *Klebsiella pneumoniae* (n = 25, 23.8%), Candida species (n = 13, 12.4%, with 11 cases of nonalbicans), Enterococcus spp. (n = 8, 7.6%), *Acinetobacter baumannii* (n = 9, 8.5%), *Enterobacter aerogenes* (n = 3, 2.8%), *Proteus mirabilis* (n = 2, 1.9%), and *Pseudomonas aeruginosa* (n = 2, 1.9%). In this study, 50.4% of organisms were MDR [16].

In a study conducted in Poland between 2016 and 2018, eight *Klebsiella pneumoniae* NDM (New Delhi metallo-beta-lactamase)-producer isolates were identified from seven patients, including two kidney transplant recipients, with urinary tract infections (UTIs) (three isolates), asymptomatic bacteriuria (ABU) (three isolates), or gut colonization [17].

UTIs associated with carbapenem-resistant *Klebsiella pneumoniae* (CRKP) seem to be related to a long length of stay and increased mortality when compared to UTIs caused by carbapenem-susceptible *K. pneumoniae* [18,19].

It is estimated that the rate of pre-transplant colonization by methicillin-resistant *Staphylococcus aureus* (MRSA) is about 10%, and therefore, transplant recipients are at high risk for MRSA infection due to predisposing factors, such as immunosuppressive therapies, surgical procedures, and ICU stay [20].

## 4. Risk Factors

Risk factors for UTIs in kidney transplant recipients can be classified into preoperative, intraoperative, and postoperative factors. The first category includes female gender, old age, diabetes mellitus, pre-transplant UTIs, urinary tract abnormalities, and prolonged dialysis. With regard to intraoperative factors, the risk of UTIs appears to be higher if the donor is deceased, if double-J ureteral stent or duplex ureters are used and in the case of prolonged use of indwelling urinary catheterization and retransplantation. In the postoperative period, the factors that contribute to UTI development are immunosuppression, graft dysfunction, rejection, and invasive urological procedures [21] (Table 1).

Female gender represents an independent risk factor, as also reported in the RESISTRA cohort [9,14]. Also, Nihan Tekkarışmaz et al. identified as risk factors for UTIs after kidney transplant: female sex (*p* = 0.001) (probably due to anatomic factors, such as the short length of the urethra and proximity of the vagina to anus) [16]. Although the female gender is considered a risk factor for UTIs in the post-transplant period [2,22], in some studies, no difference was found [23].

Regarding the difference in terms of age, the data in the literature are conflicting. Chaung et al. reported that 55% of patients >60 years old developed UTIs compared to only 30% in the younger group [24]. By contrast, other studies did not report a significant association between age and UTI incidence [25].

Nihan Tekkarışmaz et al. identified as risk factors for UTIs after kidney transplant. In addition to the female sex, the presence of pretransplant diabetes (*p* = 0.05) and diabetes were also significant risk factors for UTI recurrence [16].

Many studies reported that azathioprine, mycophenolate mofetil [24], and anti-thymocyte globulin [26] are associated with a higher incidence of UTIs, while other immunosuppressive drugs (such as calcineurin inhibitors, everolimus) have no particular impact on the risk [27,28].

In a study that evaluated basiliximab and Anti-Thymocyte Globulin (ATG) as induction therapy, more episodes of UTI occurred in the ATG-induced group [29]. Similarly, many other studies have suggested avoiding ATG in induction therapy. Moreover, immunosuppressive therapies such as cyclosporine, sirolimus, prednisolone, and mycophenolate mofetil have been reported to increase the risk of UTI [24,26]. In the study by Nihan Tekkarışmaz et al., there was no difference between induction drugs for the development of UTIs. Moreover, in the first year after transplantation, 129 patients received calcineurin inhibitors and 16 patients received mTOR inhibitors, and there was no statistically significant difference between the two drugs in terms of UTI (*p* = 0.43). Therefore, the type of induction regimen (*p* = 0.68) and the type of immunosuppressive treatment (*p* = 0.59) did not appear to be significant risk factors. However, a high daily dose of mTOR inhibitor seems to be associated with an increased risk for UTI development. Therefore, it would be better to use mTOR inhibitors at the lowest possible dose [16].

In a study, the administration of tacrolimus has been described as a risk factor for urosepsis [30].

Many studies have reported that the presence of ureteral stents can increase the risk of UTI development [31,32]. According to some studies, the increased duration of a stent in the ureter is associated with UTI development [33], while in other studies, no association was found.

Gołębiewska et al., using multivariate analysis, found urine flow impairment (such as urogenital surgery history, lower urinary tract malformations, and VUR) to be the only independent risk factor for upper UTI [15].

A recent meta-analysis of 13 studies showed that more than one-third of patients had at least one episode of UTI after kidney transplantation. The main risk factors highlighted in this meta-analysis were female gender, older age, long duration of catheterization, acute rejection, and cadaveric donors [2].

## 5. Management

### 5.1. Prophylaxis

Kidney Disease: Improving Global Outcomes (KDIGO) guidelines suggest prophylaxis with trimethoprim–sulfamethoxazole (TMP-SMX) for at least 6 months after transplantation in order to prevent UTIs and opportunistic infections (such as *Pneumocystis jirovecii*) [34].

A systematic review and meta-analysis of randomized controlled trials including 545 patients showed that prophylaxis with TMP-SMX significantly reduced the risk of sepsis by 87% and bacteriuria by 60%, without differences in graft loss and mortality. TMP-SMX was administered at the dosage of 160 + 800 mg orally daily [35].

As alternative regimens, if TMP-SMX cannot be used, although there are limited data, nitrofurantoin (contraindicated if GFR < 30 mL/min), cephalexin, or fluoroquinolones can be considered, but it has to be highlighted that primary prophylaxis for UTIs with antibiotics other than TMP-SMX has to be limited to the first month after transplantation. It should also be considered that, according to recent data about nitrofurantoin, the safety and efficacy appear to be similar to other therapeutic options in older adults with eGFR 30–60 mL/min137-13 [19,34].

Moreover, some non-pharmacological measures to reduce the risk of infection should also be considered. For example, the Guidelines of the American Society of Transplantation Infectious Diseases Community of Practice suggest limiting the duration of catheters and stents to within 4 weeks of transplantation. If a severe UTI is observed within the first 2–4 weeks after transplantation, it is appropriate to consider early stent removal, assessing the benefit–risk ratio in terms of urological complications [19].

With regard to secondary prophylaxis, in case of recurrent urinary tract infections, the use of fosfomycin orally at a dosage of 3 g once a week for a period of three to six months could be a therapeutic option. [3,19,35].

### 5.2. Treatment

#### 5.2.1. Asymptomatic Bacteriuria

Routine treatment of asymptomatic bacteriuria (AB) in KT recipients is not recommended because it is not supported by clear evidence of benefit. In fact, even if rare studies reported an association of AB with pyelonephritis or bacteremia caused by the same uropathogen, most studies did not find a strong association.

According to the 2019 Guidelines from the American Society of Transplantation Infectious Diseases Community of Practice, if asymptomatic bacteriuria in KT recipients during the post-transplant period is observed, a second urine culture should be performed in order to decide whether or not to treat. There is no recommendation to treat patients who show clearance of the initial bacteriuria or develop a different microorganism. By contrast, there is a weak recommendation to treat persistent AB with an unexplained rise in creatinine. There is a strong recommendation against the treatment of multi-drug resistant AB in order to avoid inducing further antibiotic resistance [12,19,34].

With regard to asymptomatic candiduria, some evidence suggests discouraging the therapeutic approach unless the patient is neutropenic or undergoing a urological procedure [36]. The preferred therapeutic option is fluconazole, 200–400 mg orally per day for 14 days, and adjustment of calcineurin inhibitors may be required [36]. 

While it is clear that asymptomatic bacteriuria should not be treated two months after transplantation, with all companies and guidelines advising against and not recommending screening and treating asymptomatic bacteriuria starting two months after transplantation, however, the management of asymptomatic bacteriuria in the first two months after kidney transplantation still needs to be clarified [12,34]. In fact, the methods of managing asymptomatic bacteriuria in the first two months after transplantation are unclear, and the possibility of treatment is much debated. In fact, there is a proposal to treat asymptomatic bacteriuria in these cases when a bacterial load > 10^5^ CFU is found. This indication is made and is due to a greater risk of masking symptoms in the first months after transplantation and the impact of immunosuppressive therapy in the first two months. However, further in-depth studies are needed to better understand and manage this condition [37,38,39].

#### 5.2.2. Cystitis

For outpatients with simple cystitis without fever, leukocytosis, or graft tenderness, the main therapeutic strategies include an oral fluoroquinolone, amoxicillin–clavulanate, or an oral third-generation cephalosporin (such as cefixime), as an empiric regimen. Nitrofurantoin should also be taken into account, although initially contraindicated for CrCl values <60 cc/min because a re-examination of data proved its safety and efficacy if CrCl >30 cc/min. Fosfomycin is not the preferred choice, so it should be reserved for more drug-resistant cystitis. With regard to the duration of treatment, according to the Guidelines from the American Society of Transplantation Infectious Diseases Community of Practice (2019), it should be 5–10 days. According to some studies, simple cystitis in KT recipients should be treated for 5–7 days, especially if it occurs beyond the early post-transplant period. By contrast, according to other studies, if the UTIs occur in the early post-transplant period (e.g., in the first 6 months), the duration of treatment should be 7–10 days [12].

Other studies show that the timing of transplantation should be taken into account for the definition of the duration of therapy (10–14 days in the early post-transplant period, 5–7 days after 6 months) [4], adjusting the dosage according to the graft function.

#### 5.2.3. Pyelonephritis and Other Complicated UTIs

In the suspicion of pyelonephritis (APN) or other complications, hospitalization to set up an intravenous therapy with an active spectrum on both Gram-negative and Gram-positive organisms (such as piperacillin–tazobactam 4.5 g IV every 6 h, meropenem 1 g IV every 8 h, cefepime 1 g IV every 8 h), is required [34]. For stable patients with mild pyelonephritis, ceftriaxone, ampicillin–sulbactam, or ciprofloxacin can represent possible therapeutic agents. In severe infections such as septic shock, reduction or discontinuation of immunosuppressive therapy should also be evaluated. Imaging is also warranted to exclude the involvement of the upper urinary tract, such as abscesses, emphysematous pyelonephritis, or urinary obstruction. All patients with complicated UTIs should be treated for 14–21 days; treatment should be continued until adequate drainage of abscesses is performed [12].

A delicate issue to address concerns the management of the critical patient and, in particular, the patient in sepsis or septic shock [12,34]. In this regard, once the diagnostic process has been started, it is necessary to consider an empirical therapy approach [12,34]. To this purpose, it is essential to be aware of the anamnesis and microbiological history of the patient in question. In particular, the approach with empirical therapy should be based on the awareness of the patient’s possible previous isolates, taking into account that the category of kidney transplant recipients often has a history of colonization by *Enterobacteriaceae* ESBL [40,41]. In fact, in the face of such scenarios, the possibility of empirically using third and fourth-generation cephalosporins falls a bit. The use of piperacillin/tazobactam is also debated since this combination appears sub-optima in the case of ESBL-producing germs [40,41]. At the same time, the use of quinolones is increasingly severely restricted due to increasing resistance rates, while cotrimoxazole has limited possibilities of use due to the potential impact on renal function [42,43].

Therefore, in this setting, the therapeutic choice would orient towards the use of carbapenems in order to guarantee the efficacy and safety of the patient. However, this choice must be weighed in relation to the severity of the patient’s clinical picture and the possibilities of waiting times for microbiological investigations [12,34]. In fact, if the clinical picture is stable and the patient does not present severity criteria, understood as a picture of sepsis or septic shock, the empirical antibiotic therapy approach could aim to save carbapenems and evaluate the possible use of piperacillin/tazobactam. In the case of the severity of the clinical picture and taking into account the risk of MDR germs, in this case, the empirical therapeutic approach would involve the use of drugs with a broader spectrum [40,41,42,43] (Table 2).

Furthermore, although *Enterobacteriaceae* ESBLs appear to be more prevalent, the potential etiological roles of *Pseudomonas aeruginosa* and *Enterococci* spp. should not be overlooked. Such awareness is essential in the empirical therapeutic approach in the face of a condition of clinical severity [34,44].

#### 5.2.4. Antimicrobial Resistance

Carbapenems are the main treatment for MDR *Enterobacteriaceae* because they often show resistance to quinolones and cotrimoxazole [40].

For extremely drug-resistant Gram-negative bacteria such as *Pseudomonas* spp. and *Klebsiella* spp., there are new therapeutic options represented by ceftolozane–tazobactam, ceftazidime–avibactam, meropenem–vaborbactam, imipenem–cilastatin–relebactam, aztreonam–avibactam and cefiderocol which have proven to be valid alternatives to colistin and aminoglycosides. Nevertheless, emerging resistance against these agents has already been found [41,42]. Nevertheless, their use depends on the stage of renal dysfunction, and individual dosing is required [43] (Table 2).

Fosfomycin can be evaluated as an alternative agent for the treatment of MDR UTIs, especially if limited to cystitis. Multiple doses of oral fosfomycin scheduled 48–72 h are preferred, which seems to be more effective in this setting of transplanted patients, rather than the single dose of 3 g, which is associated with higher failure rates. Oral pivmecillinam seems to be active against ESBL-producing *E. coli* as a treatment for cystitis. Few data about minocycline and doxycycline in the population of kidney transplant recipients are reported in the literature [12].

The treatment of methicillin-resistant *Staphylococcus aureus* (MRSA) and vancomycin-resistant enterococcus (VRE) infections remains a difficult challenge. This underlines the importance of screening for MRSA before the kidney transplant through nasal swabs, with possible decolonization procedure of carriers by intranasal application of 2% topical mupirocin twice daily for 5 days and chlorhexidine baths for 7 days [44] (Table 2).

#### 5.2.5. Recurrent UTIs

In the case of recurrent UTIs in kidney transplant recipients, defined as ≥2 UTIs in a 6-month period or ≥3 UTIs in a 12-month period, it is mandatory to look for predisposing factors in order to adopt the most appropriate strategies to correct them, including lifestyle modification (such as hydration, frequent voiding, voiding after sexual intercourse), non-antimicrobial prevention (such as topical vaginal estrogen for post-menopausal women, lactobacillus-containing probiotics, methenamine hippurate, l-methionine, and cranberry). However, few and conflicting data are available about non-antimicrobial strategies in KT recipients. Therefore, imaging should be performed to identify obstruction, renal calculi, foreign bodies, complex cysts, and other structural factors. Other diagnostic examinations, such as voiding cystourethrograms, urodynamic studies, and cystoscopy, should be taken into account. With regard to vesicoureteral reflux (VUR), there is one retrospective study that evaluated the recurrence of UTIs after VUR surgical correction in 60 KT recipients with recurrent febrile UTIs. The results showed a decrease in UTI recurrence from a median of four episodes in a year before surgery to a median of one episode in a year after surgery (*p* < 0.05) [45].

One of the most important risk factors for recurrent UTI is infection with MDR pathogens, especially with *K. pneumoniae* [46].

In case of recurrent urinary tract infections, the possibility of secondary prophylaxis with 3 g of fosfomycin once a week orally for 3–6 months may be considered [47].

#### 5.2.6. Donor-Derived Infections

When donor infection/colonization is found on blood or urine samples collected before the donation, a prophylactic antimicrobial regimen in the recipient is recommended, with a duration of 14 days if the identified microorganisms are Gram-negative bacilli (especially *Pseudomonas*), *Staphylococcus aureus* or *Candida* spp. and with a duration of 7 days if less virulent pathogens are found [12].

## 6. Short and Long-Term Outcomes

The presence of a UTI in the first year post-transplantation has been reported to increase allograft loss by 29% and mortality by 41% [48]. Moreover, if untreated, a UTI occurring during the first 3 months after transplant can increase the risk of acute rejection [22].

The possible graft deterioration after UTI finds its explanation on the one hand in the elicitation by the pathogen of the immune system with subsequent acute or chronic rejection and on the other hand in the interstitial scars that may develop as a result of acute graft pyelonephritis [49]. Possible causes of graft deterioration include the reduction/discontinuation of immunosuppressive therapy following the diagnosis of acute graft pyelonephritis.

Notably, a study conducted by Rice et al. reported an association between upper UTIs by virulent *E. coli* and acute allograft injury [50].

Deterioration of renal function of the graft after UTI episodes was also assessed in the study by Nihan Tekkarışmaz et al., and the mean eGFR 1 year after transplantation significantly decreased in the UTI group (*p* = 0.006).

Post-transplant APN can affect graft function and increase mortality [5].

In a large cohort study conducted by Abbott et al., the adjusted relative risk for graft failure in recipients with late UTIs was 2.35 times higher when compared with patients without UTIs [3].

A retrospective study that enrolled 380 patients showed that the presence of recurrent UTIs during the first year after transplantation was associated with worse graft outcomes (eGFR value < 60 mL/min/1.73 m^2^) [51].

A Spanish study reported that acute graft pyelonephritis was significantly associated with graft function impairment and with a higher risk of graft loss one year after transplantation, while lower UTIs did not seem to have an impact on graft function [46].

Moreover, one study that included a large cohort of kidney transplant recipients in the United States reported an association between late UTIs and an increased risk of death [3].

Similarly, a study conducted by Britt and colleagues reported poorer graft survival in recipients with recurrent UTIs when compared to those without UTIs or with non-recurrent UTIs [52].

By contrast, other studies did not find an association between survival or graft damage and recurrence of UTIs [8].

## 7. Other Management Perspectives

Fecal microbiota transplantation (FMT) from healthy donors is an unconventional solution for KTx recipients with recurrent UTIs. Grosen et al. demonstrated that FMT might be an effective way to prevent recurrent UTIs with *K. pneumoniae* ESBL+ in KTx recipients. No multi-drug-resistant *K. pneumoniae* bacteria were identified in urine or feces for 12 months [53].

Bacteriophage therapy can also be a potential alternative against MDR urinary tract infections [54]. Bacteriophages are viruses that attack bacterial cells, causing lysis of the host (lytic lifestyles) or inserting into the bacterial genome, existing as a prophage (lysogenic state). Many studies report diverse lytic bacteriophages to *K. pneumoniae* and their potential in vitro. According to Ujmajuridze et al., bacteriophage therapy might also be effective and safe for treating UTIs [55]. Phages can be administered alone (selected doses of phages or phage cocktail) or together with antibiotics or disinfectants, observing the synergism of action for this combination [56]. So far, single case reports have been published showing the successful use of phage therapy in the treatment of relapsing ESBL-producing *K. pneumoniae* UTIs in KTx recipients [57]. Studies by the Polish team investigating phage therapy showed both treatment success and improvement of kidney allograft function in KTx recipients with complicated UTIs [58]. This indicates the high potential of phage therapy in urology and nephrology, including KTx recipients [54,55,56,57,58].

## 8. Conclusions

Our review highlights how urinary tract infections in kidney recipients are currently an open challenge. In particular, although from an epidemiological and microbiological point of view, Enterobacteriaceae constitute and remain the prevalent etiological agents, MDR germs are taking on an increasingly important role. The burden of antimicrobial resistance is increasingly impacting the evolution and management of this infection. Several risk factors have been identified, and their recognition is essential in order to intervene as preventively as possible. After all, the potential benefits and harm of secondary prophylaxis must be balanced in order to impact antimicrobial resistance as little as possible. Areas of uncertainty include also the management of asymptomatic bacteriuria in the early stages of post-transplantation.

In the face of such scenarios, urinary tract infections in kidney transplant patients are one of the main complications both in terms of graft survival and in terms of transplant recipient survival. For this reason, the management of such conditions requires a multidisciplinary approach and collaboration involving various specialists, such as infectious disease specialists, nephrologists, urologists, and transplant surgeons. Such management is fundamental and essential in order to best address such conditions.

## Figures and Tables

**Table 1 microorganisms-12-02217-t001:** Risk factors.

Preoperative	Intraoperative	Postoperative
Female gender	Deceased donor	Immunosuppression
Old age	Double-J ureteral stents	Graft dysfunction
Diabetes mellitus	Prolonged indwelling urinary catheterization	Graft rejection
Pre-transplant UTIs	Retransplantation	Invasive urological procedures
Urinary tract abnormalities		
Prolonged dialysis		

**Table 2 microorganisms-12-02217-t002:** Microbiologic agents and potential antibiotic therapies.

Microbiological Agents	Resistance Mechanisms	Potential Antibiotic Therapies
*Enterobacteriaceae*	AmpC	Cefepime
		Ceftolozane/Tazobactam
	ESBLs	Ertapenem/Meropenem
		Ceftolozane/Tazobactam
	KPC	Ceftazidime/Avibactam
		Meropenem/Vaborbactam
		Imipenem/Cilastatin/Relebactam Cefiderocol
	OXA-48	Ceftazidime/Avibactam
		Cefiderocol
	IMP, VIM, NDM	Aztreonam/Avibactam
		Cefiderocol
*Pseudomonas aeruginosa*	MDR	Meropenem
		Ceftolozane/Tazobactam
	XDR CP	Ceftazidime/Avibactam
		Imipenem/Cilastatin/Relebactam
	MC	Aztreonam/Avibactam
		Cefiderocol
*Enterococci* spp.	Ampi-R	Vancomycin
	VRE	Teicoplanin
	Van-A	Linezolid
		Tigecycline
	Van-B	Linezolid
		Daptomycin
		Tigecycline
		Teicoplanin

AmpC: ampicillinase class C, ESBLs: Extended-Spectrum β-Lactamases, KPC: *Klebsiella Pneumoniae* Carbapenemase, OXA-48: Oxacillinase-48, IMP: Imipenemase, VIM: Verona-Intergon-encoded Metallo-β-lactamase, NDM: New Delhi Metallo-β-lactamase, Ampi-R:, MDR: Multi-drug Resistance, XDR: Extensively Drug-Resistant, CP: Carbapenemase Producing, MC: Metallo-Carbapenemase, VRE: Vancomycin Resistant Enterococci, Van-A: Vancomycin-A, Van-B:Vancomycin-B.

## Data Availability

It is possible to request the data directly from the corresponding author via email.

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
