# Peer review of "Urinary Tract Infections in Kidney Transplant Patients: An Open Challenge—Update on Epidemiology, Risk Factors and Management"

_microorganisms, 2024, doi:10.3390/microorganisms12112217_

Round 1
Reviewer 1 Report
Comments and Suggestions for Authors
Biagio Pinchera et al present a review of the literature around Urinary Tract Infections in kidney transplant patients, the epidemiology, risk factors and management. The topic of the manuscript is of interest given that up to 74% of kidney transplant patients experience at least one episode of UTIs in the first year after transplantation, increasing the risk of graft loss and mortality. In addition, authors provide some evidence on the impact of the increasing antimicrobial resistance. Tables are clear and give a summary of a variety of key concepts. The content of the manuscript is well structured and clearly presents the information in a meaningful way to the reader.
The literature reviewed is up to date and the manuscript follows a logical order, which makes it easy to read. Overall, the manuscript is detailed, well written and summarizes the current state of play.
I just have some minor points, which are detailed below and should be addressed before publication.
1- Page 2, Microbiology: It would be helpful to organize this information on a Table.
2- Page 3, second paragraph: please, define NDM.
3- Page 4, third paragraph: please, define ATG.
4- Page 4, Prophylaxis: please, define KDIGO
5- Page 7, Table 2, “Ceftolozane/Tazobactam*”: there is an asterisk that was not defined.
6- Page 8: “although few and conflicting data are available about non-antimicrobial strategies in KT recipents, antimicrobial intervention, structural factors correction.” Please, check this statement since it seems that there is something missing.
Author Response
Response to Reviewer 1
First, I thank the Reviewer 1 for the comments, reviews and suggestions. Your contribution and support have been fundamental and essential for the improvement of the manuscript.
I proceeded to carry out all the indicated and suggested revisions. Below are the specific revisions made, as also highlighted in the manuscript.
Response to Reviewer 1
1- Page 2, Microbiology: It would be helpful to organize this information on a Table.
I thank you for the suggestion and the advice, however, having already included two tables, we are
unable to add a third table.
2- Page 3, second paragraph: please, define NDM.
I have carried out the indicated revision, as reported in the manuscript.
3- Page 4, third paragraph: please, define ATG.
I have carried out the indicated revision, as reported in the manuscript.
4- Page 4, Prophylaxis: please, define KDIGO
I have carried out the indicated revision, as reported in the manuscript.
5- Page 7, Table 2, “Ceftolozane/Tazobactam*”: there is an asterisk that was not defined.
I have corrected it, as reported in the manuscript.
6- Page 8: “although few and conflicting data are available about non-antimicrobial strategies in KT recipents, antimicrobial intervention, structural factors correction.” Please, check this statement since it seems that there is something missing.
I have corrected it, as reported in the manuscript.
I thank the Reviewer 1 for the fundamental contribution and essential support.
Thank you.

Reviewer 2 Report
Comments and Suggestions for Authors
Dear Authors,
Thank you very much for this comprehensive review concerning important problem of UTI among transplanted patients. I have no notes , manuscript is ready for publishing
Regards
Reviewer
Author Response
Responses to Reviewer 2
First, I thank the Reviewer 2 for the comments, reviews and suggestions. Your contribution and support have been fundamental and essential for the improvement of the manuscript.
I proceeded to carry out all the indicated and suggested revisions. Below are the specific revisions made, as also highlighted in the manuscript.
I thank the Editor and the Reviewers for the fundamental contribution and essential support.
Thank you.
